# Targeted deletion of AKAP7 in dentate granule cells impairs spatial discrimination

Brian W Jones[1‡], Jennifer Deem[1‡], Thomas J Younts[2‡], Michael Weisenhaus[1], Christina A Sanford[1], Margaret C Slack[1], Jenesa Chin[1], Daniela Nachmanson[1], Alex McKennon[1], Pablo E Castillo[2], G Stanley McKnight[1]*

[1]Department of Pharmacology, University of Washington School of Medicine, Seattle, United States; [2]Dominick P Purpura Department of Neuroscience, Albert Einstein College of Medicine, New York, United States

**Abstract** Protein Kinase A (PKA) mediates synaptic plasticity and is widely implicated in learning and memory. The hippocampal dentate gyrus (DG) is thought to be responsible for processing and encoding distinct contextual associations in response to highly similar inputs. The mossy fiber (MF) axons of the dentate granule cells convey strong excitatory drive to CA3 pyramidal neurons and express presynaptic, PKA-dependent forms of plasticity. Here, we demonstrate an essential role for the PKA anchoring protein, AKAP7, in mouse MF axons and terminals. Genetic ablation of AKAP7 specifically from dentate granule cells results in disruption of MF-CA3 LTP directly initiated by cAMP, and the AKAP7 mutant mice are selectively deficient in pattern separation behaviors. Our results suggest that the AKAP7/PKA complex in the MF projections plays an essential role in synaptic plasticity and contextual memory formation.

*For correspondence: mcknight@ u.washington.edu

‡These authors contributed equally to this work

Competing interests: The authors declare that no competing interests exist.

## Introduction

The hippocampal formation, which comprises the hippocampus and dentate gyrus (DG), plays a crucial role in the encoding and retrieval of episodic and spatial memories (*Burgess et al., 2002*). The DG receives input from the entorhinal cortex and sends its output via dentate granule cell (DGC) axons or mossy fibers (MFs) to synapse with proximal dendrites of hippocampal CA3 pyramidal neurons (*Amaral et al., 2007*). Previous theoretical studies predicted that one function of the DG is to accurately separate similar memories by orthogonally encoding discrete non-overlapping input patterns onto the CA3 field (*Lee and Solivan, 2010*; *Rolls, 2013*; *Schmidt et al., 2012*). This precise form of encoding is known as pattern separation (*Gilbert et al., 2001*; *Kesner and Rolls, 2015*; *Leutgeb et al., 2007*; *Nakashiba et al., 2012*). Behavioral studies have helped to clarify the role of the circuit between the DG and region CA3 in modulating spatial and contextual pattern separation (*Aimone et al., 2011*; *Myers and Scharfman, 2011*; *Treves et al., 2008*). However, the cellular and molecular mechanisms that contribute to this functional role in pattern separation are incompletely understood.

Activity-dependent, long-term changes in synaptic transmission (e.g. long-term potentiation, LTP, and depression, LTD) within hippocampal circuits provide a basis for learning and memory. Protein Kinase A (PKA) is a key mediator of both pre- and postsynaptic forms of long-term plasticity (*Kandel et al., 2014*). A Kinase Anchoring Proteins (AKAPs) are multi-domain scaffolding proteins that localize PKA and other proteins to discrete subcellular domains and spatially restrict intracellular signaling events that are required for neuroplasticity (*Dell'Acqua et al., 2006*; *Scott et al., 2013*). We, and others, have shown that genetic deletion of a dendritic AKAP (AKAP5) results in delocalization of PKA from the postsynaptic dendrites of CA1 and CA3 hippocampal neurons and leads to behavioral defects in memory and learning (*Lu et al., 2007*; *Tunquist et al., 2008*;

*Weisenhaus et al., 2010*). Previous studies have shown localization of PKA-RIIβ to MF projections (*Glantz et al., 1992*; *Weisenhaus et al., 2010*) suggesting the presence of a presynaptic anchoring protein. However, no presynaptic AKAP has been previously reported in adult neurons. Moreover, a role for AKAPs in pattern separation has not been explored.

The MF-CA3 synapse is well-known for expressing a form of presynaptic, PKA-dependent, and NMDA receptor-independent LTP (herein referred to as MF-LTP) (*Nicoll and Schmitz, 2005*). The molecular nature of this presynaptic LTP is not fully understood, although several protein targets have been identified (*Castillo, 2012*). A model for MF-LTP suggests that presynaptic calcium influx activates calcium-sensitive adenylyl cyclases (ACs) that produce cyclic AMP (cAMP) leading to PKA activation and the phosphorylation of substrates (*Huang et al., 1994*; *Villacres et al., 1998*; *Wang et al., 2003*; *Weisskopf et al., 1994*). MF-LTP can be induced either synaptically by repetitive stimulation of MFs (tetanus) or by direct elevation of cAMP by the application of the adenylyl cyclase (AC) activator, forskolin. Previous studies using mice with defects in specific cyclases or subunits of PKA have suggested mechanistic differences in synaptic versus forkolin-induced LTP in this pathway (*Huang et al., 1995*; *Wong et al., 1999*). The role of sub-cellular localization of the cAMP/PKA signaling pathway has not been addressed in DGCs and could play a critical role in both synaptic plasticity and contextual discrimination.

We report that AKAP7 (also referred to as AKAP15/18) is expressed in many regions of the brain including the hippocampus. In the hippocampal formation, AKAP7 is selectively expressed in DGCs and localizes to both dendrites and the MF presynaptic projections. Loss of AKAP7 results in delocalization of PKA-RIIβ from the MF projections. AKAP7 global KO mice demonstrate deficits in non-cued spatial and contextual pattern separation, behaviors that have been associated with DG function. Moreover, we deleted AKAP7 from DGCs and found that the behavioral deficits seen in the global KO are specific to the DG-CA3 pathway. Slice electrophysiology revealed that AKAP7 deficient animals were severely impaired in cAMP-induced MF-LTP. Our results identify AKAP7 as a unique AKAP that localizes PKA to axons and terminals of adult neurons and we demonstrate that AKAP7 functions to regulate both synaptic plasticity and behavior.

## Results

### AKAP7 is highly expressed in dentate granule cells

The *Akap7* gene encodes multiple isoforms using two separate promoters to generate either long (γ/δ) or short (α) isoforms (*Figure 1A*). Only the final exon, containing the PKA-binding domain characteristic of AKAPs, is shared by all AKAP7 isoforms. The final exon also contains a modified leucine zipper domain which interacts with brain $Na^+$ and L-type $Ca^{2+}$ channels and facilitates PKA modulation of ion conductance through these channels (*Marshall et al., 2011*; *Tibbs et al., 1998*). Long and short AKAP7 isoforms differ in subcellular localization. Long isoforms, γ/δ, are predominantly cytosolic, although they contain a potential nuclear localization sequence. Only the short α isoform contains N-terminal myristoylation and palmitoylation sites, facilitating plasma membrane association (*Gray et al., 1998*). The long isoforms are expressed in all tissues examined including brain but the short isoform is predominantly expressed in neurons (*Jones et al., 2012*) and found in many regions of the brain including the hippocampus, cortex, striatum, thalamus and cerebellum (*Figure 1—figure supplement 1*). Western immunoblotting demonstrated that the α isoform is much more abundant than γ/δ in the hippocampus (*Figure 1B*) as well as other brain regions (*Figure 1—figure supplement 1*). We used RiboTag polyribosome immunoprecipitation (*Sanz et al., 2009*) to isolate DGC-specific polysome-associated mRNA from hippocampal homogenates to demonstrate that DGCs specifically express AKAP7α and not AKAP7γ/δ (*Figure 1C* and *Figure 1—figure supplement 2*).

We previously reported the insertion of loxP sites surrounding the last exon of the *Akap7* gene (as shown in *Figure 1A*) and the use of *Meox2^Cre* mice to produce a global AKAP7 KO (*Jones et al., 2012*). Western blot analysis of hippocampal extracts from WT and AKAP7 KO mice demonstrate the complete loss of both the long and short isoforms of AKAP7 with no change in the expression level of RIIβ (*Figure 1B*). Immunohistochemistry (*Figure 1D*) revealed the specific expression of AKAP7 in WT DGCs, the localization of AKAP7 protein to the MF projections and dendrites, and the complete loss of AKAP7 expression in the KO. The global KO of AKAP7 did not disrupt the overall

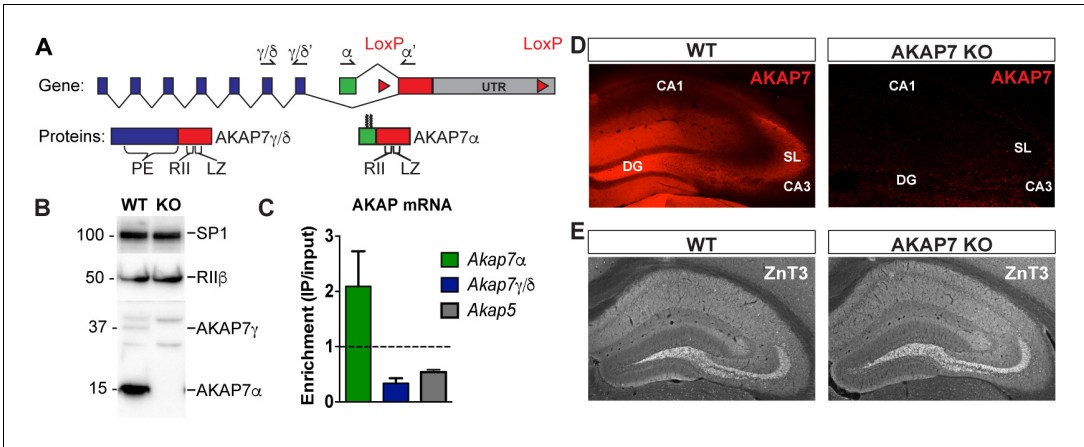

**Figure 1.** AKAP7 is highly expressed in the dentate gyrus. (**A**) Map of *Akap7* gene (introns not to scale) showing engineered loxP sites and location of primer sets to distinguish between long and short isoforms. AKAP7 proteins resulting from alternate splicing showing phosphoesterase (PE), PKA-RII–binding (RII), and modified leucine zipper (LZ) domains and N-terminal lipid modifications on AKAP7α. (**B**) Immunoblot of hippocampal lysates shows that the AKAP7α protein is expressed at much higher levels than AKAP7γ. The protein expression of PKA-RIIβ is not reduced in AKAP7 KO hippocampus. Transcription factor SP1 was used as a loading control. Immunoblots were done using the hippocampus from one animal and replicated three or more times. (**C**) RiboTag analysis of DG-specific transcripts compared to total hippocampal transcripts shows enrichment for AKAP7α and de-enrichment of AKAP7γ/δ and AKAP5. (**D**) Immunofluorescence microscopy of WT or KO hippocampus with an antibody against AKAP7 reveals expression in DGCs but not areas CA1 or CA3. (**E**) Immunofluorescence microscopy with an antibody against the presynaptic zinc transporter-3 (ZNT3) shows normal morphology of the mossy fiber field and hippocampal layering in KO compared to WT. Images are representative of multiple slices from ≥3 mice.

The following figure supplements are available for figure 1:

**Figure supplement 1.** AKAP7 is widely expressed throughout the brain and in some regions appears restricted to specific neurons.

**Figure supplement 2.** Validation of DG-specificity of RiboTag Immunoprecipitation.

structure of the hippocampus and the MF projections remain intact as shown by immunohistochemistry for zinc transporter-3 (ZNT3), a marker for MF terminals (*Figure 1E*). Our results demonstrate that in the hippocampus, the short form of AKAP7 (AKAP7α) is preferentially expressed in DGCs and localizes to both dendrites and the MF axonal projections.

## AKAP7 anchors PKA in MF axons and terminals

Within the hippocampus, the stratum lucidum is a visually distinct layer containing unmyelinated MF axons, large presynaptic MF boutons, and their postsynaptic contacts from CA3 pyramidal neurons. We colocalized AKAP7 staining with pre- and postsynaptic markers. AKAP7 co-localized with neurofilament-M (NF-M), a protein found in axons, and with ZNT3 and synaptophysin (SYP), which are present in MF boutons. We saw no co-localization of AKAP7 with markers for the post-synaptic density, post-synaptic density-95 (PSD-95), or dendrites, microtubule associated protein-2 (MAP-2), indicating that AKAP7 is localized in the MFs and not the CA3 dendrites (*Figure 2*). Together, our data indicate that AKAP7α is localized throughout MF boutons and axons, identifying it as the only known presynaptic AKAP in mature neurons.

Deletion of AKAP7 resulted in delocalization of PKA-RIIβ from the MF projections as shown in *Figure 3A and B*. This was not due to changes in overall PKA-RIIβ protein levels (*Figure 1B*) but rather delocalization of PKA-RIIβ back to the cell body and dendrites of DGCs. This result strongly suggests that AKAP7 is the sole AKAP anchoring PKA-RIIβ in MF axons. Staining for the catalytic subunit, PKA-C, revealed that most of the C subunit was also displaced from the stratum lucidum in the AKAP7 KO and what remained did not colocalize with NF-M suggesting that the remaining C

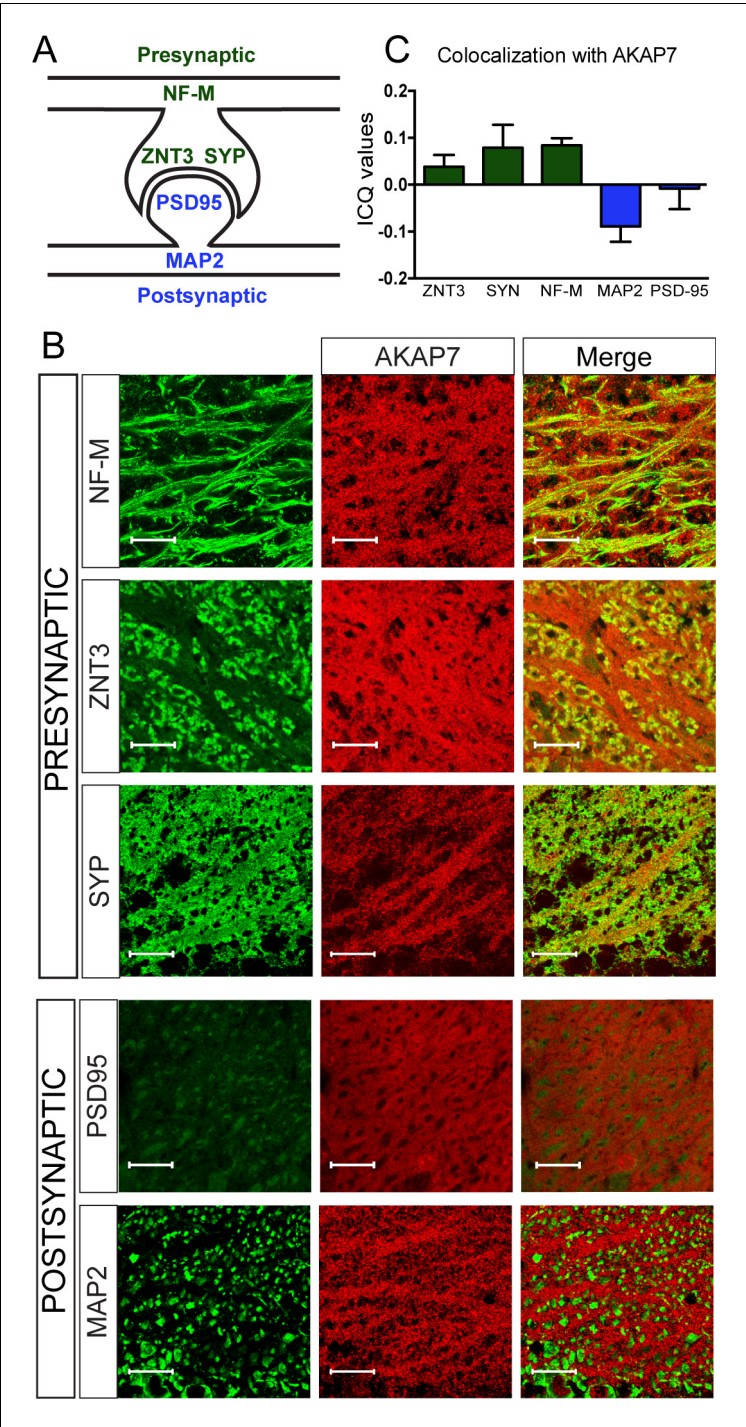

**Figure 2.** AKAP7 localizes throughout the mossy fiber axons and terminals. (**A**) Model of the localization of pre- and postsynaptic proteins. (**B**) Confocal images of stratum lucidum show AKAP7 distributed throughout mossy fiber axons and boutons but not in postsynaptic compartments. Top: AKAP7 in red, Presynaptic markers: neurofilament-M (NF-M), zinc transporter 3 (ZnT3), synaptophysin (SYP). Bottom: Post-synaptic markers: post-synaptic density protein-95 (PSD95), or microtubule-associated protein 2 (MAP2) in green. Images are representative of multiple slices from ≥3 mice. Scale bars, 20 μm. (**C**) Colocalization analysis of regions of interest within stratum lucidum using the Coloc2 plug-in from Fiji (ImageJ) as described in Materials and methods. Intensity correlation quotient (ICQ) values between 0 and 0.5 indicate some colocalization and values between 0 and −0.5 indicate segregated staining.

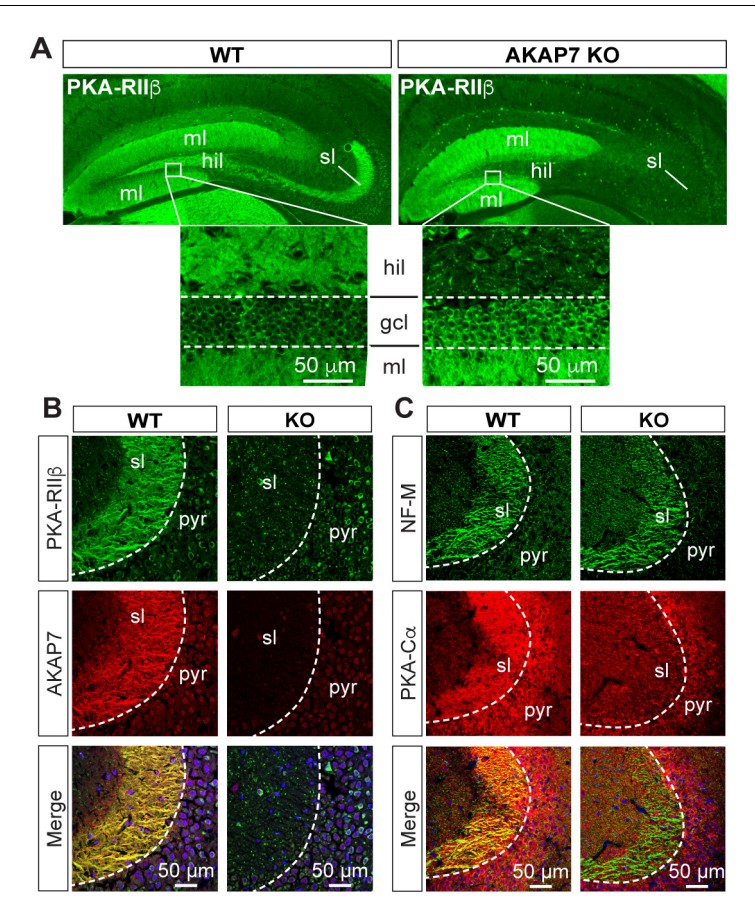

**Figure 3.** AKAP7 anchors PKA in the mossy fiber axons and terminals of DGCs. (**A**) PKA-RIIβ is lost from mossy fibers in the hillus (hil) and stratum lucidum (sl) but increased in soma (gcl) and dendrites (ml) in AKAP7 KO. (**B**). Confocal images of stratum lucidum (sl) showing PKA-RIIβ (green) colocalized with AKAP7 (red) in mossy fibers in WT but not KO animals. Nuclei are stained with DAPI (blue); CA3 pyramidal cell layer (pyr). (**C**) Confocal images as in *B* showing NF-M (green) colocalized with PKA-Cα (red) in mossy fibers in WT but not KO animals. Regions of interest within stratum lucidum were analyzed using the Coloc2 plug-in from Fiji (ImageJ) as described in Materials and methods and the analysis of PKA-Cα and NF-M gave an ICQ value of 0.13 in WT and −0.035 in the KO indicating colocalization in the WT but not in the KO. Images are representative of multiple slices from ≥three mice. Nuclei are stained with DAPI (blue) in merged images.

subunit was in CA3 pyramidal cells, interneurons, or glia (*Figure 3C*). Thus, AKAP7 is required for localization of the PKA holoenzyme to MF axons and terminals.

## Disruption of AKAP7 results in deficits in MF-CA3 LTP

The MF-CA3 synapse exhibits a form of LTP that is expressed presynaptically as increased neuro-transmitter release and is dependent on PKA activity (*Huang et al., 1994*; *Weisskopf et al., 1994*). Since the AKAP7 KO mice revealed a dramatic delocalization of PKA from MF projections, we reasoned that presynaptic PKA-dependent forms of long-term plasticity might be disrupted. To examine this possibility, we monitored MF-CA3 synaptic transmission and plasticity in acute hippocampal slices acquired from AKAP7 WT and KO mice (*Figure 4A*). Cyclic AMP-induced LTP, monitored with extracellular field excitatory postsynaptic potential (fEPSP) recordings, was nearly abolished in the AKAP7 KO mice when compared to WT littermate controls (*Figure 4B*). To more directly measure presynaptic function at the MF-CA3 synapse, we performed whole-cell recordings of NMDA receptor (NMDAR)-mediated excitatory postsynaptic currents (EPSCs) under experimental conditions designed to reduce polysynaptic contamination generated by associational-commissural inputs onto

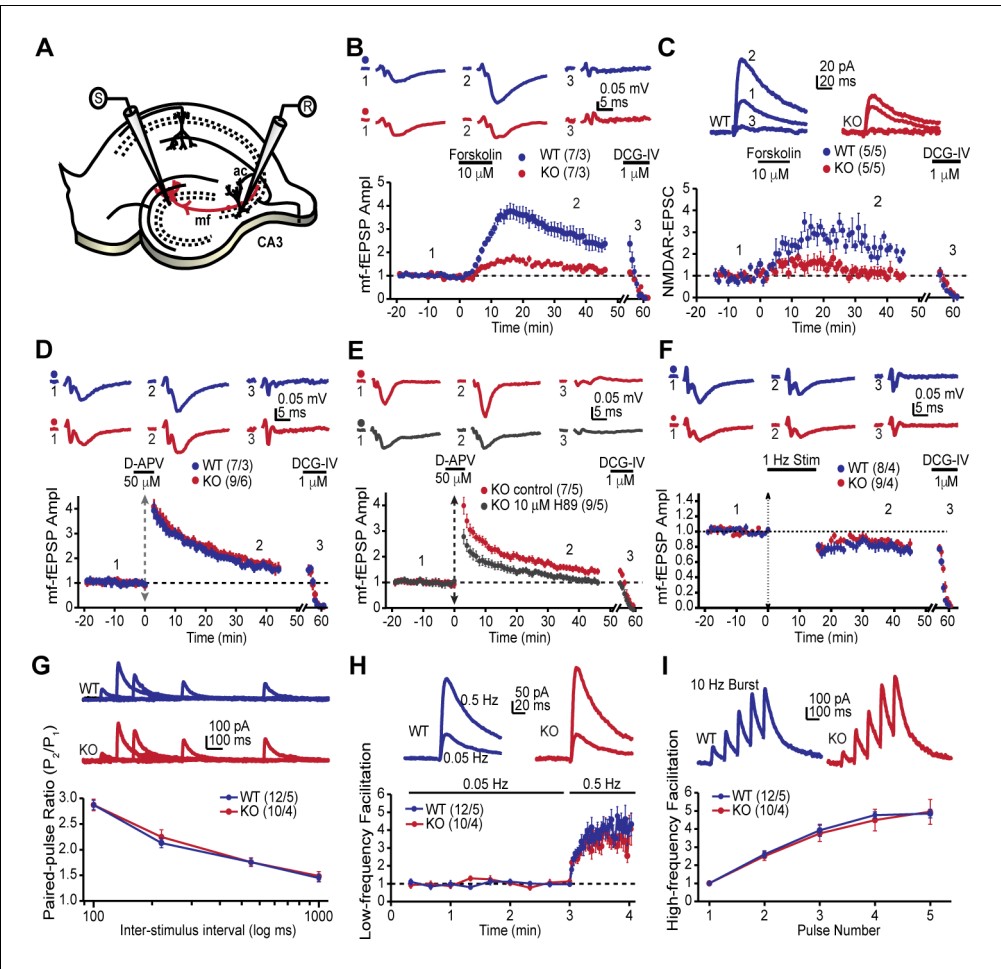

**Figure 4.** AKAP7 is required for cAMP-induced LTP at MF-CA3 pyramidal cell synapses. (**A**) Model of the placement of stimulating and recording electrodes. (**B**) cAMP-induced LTP at the MF-CA3 synapse was tested by both field EPSPs (fEPSPs), and (**C**) whole cell recordings from acutely isolated WT and AKAP7 KO brain slices. Field EPSPs (WT: 2.4 ± 0.3 of baseline vs. KO: 1.28 ± 0.07 of baseline; p=0.0029; Student's unpaired t-test; Cohen's d = 1.98). Intracellular whole-cell voltage-clamp recordings from CA3 pyramidal cells (WT: 2.13 ± 0.16 of baseline vs. KO: 1.09 ± 0.18 of baseline; p=0.0027; Student's unpaired t-test; Cohen's d = 2.70). Magnitude of LTP was assessed 35–45 min post application of forskolin to elevate cAMP. (**D**) Tetanus-induced MF-LTP was intact in AKAP7 KO slices. Experimental conditions identical to panel B, but LTP was induced with a tetanus (vertical double arrow) consisting of two bursts (each with 125 pulses at 25 Hz) spaced by 20 s. To block postsynaptic NMDA receptor-mediated MF-LTP, the NMDA receptor antagonist D-APV (50 μM, horizontal black bar) was bath applied before and washed out immediately after the tetanus. (**E**) Tetanus-induced MF-LTP was inhibited in AKAP7 KO slices pre-incubated (1 hr) and continuously perfused with the PKA inhibitor H89 (10 μM). Control: 1.48 ± 0.08 of baseline vs. H89: 1.08 ± 0.06; p=0.0013 (Student's unpaired t-test); Cohen's d = 2.00. Experimental conditions were otherwise identical to panel D. (**F**) Long-term depression at the MF-CA3 synapse was normal in AKAP7 KO mice. Long-term depression was induced (first horizontal black bar) with the classical 1 Hz, 15-min protocol. For panels B-F representative traces are shown above with numbers corresponding to the summary time course plot below the electrophysiology figures. 1 μM DCG-IV (second horizontal black bar) was bath applied at the end of each experiment to confirm identity of mossy fiber synaptic responses. Since the DCG-IV application time was slightly different between experiments, a break in the x-axis was introduced and the DCG-IV effect was time-aligned. Measures of short-term plasticity at the MF-CA3 synapse were intact in the AKAP7 KO mice compared to WT controls: basal probability of release (**G**), low-frequency facilitation (**H**), and high-frequency facilitation (**I**) were tested. Release probability was inferred from measurements of the paired-pulse ratio, calculated by dividing the second of two successive pulses by the first. Short-term plasticity was measured using whole-cell voltage clamp recordings of isolated NMDAR-EPSCs. For each group of mice the values in parentheses correspond to the number of slices/number of mice.

*Figure 4 continued on next page*

*Figure 4 continued*

The following figure supplement is available for figure 4:

**Figure supplement 1.** Whole-cell voltage clamp recordings confirm intact tetanus-induced MF-LTP.

the CA3 pyramidal cells (*Kwon and Castillo, 2008*; *Nicoll and Schmitz, 2005*). Consistent with our field recording measurements, cAMP-induced MF-LTP was severely reduced in AKAP7 KO mice compared with WT controls (*Figure 4C*).

Intriguingly, tetanus-induced LTP remained intact in the AKAP7 KO mice (*Figure 4D*). We also examined tetanus-induced LTP using whole cell recordings of NMDAR-EPSCs and found a similar result (*Figure 4—figure supplement 1*). Since tetanus-induced LTP in the MF-CA3 pathway has previously been shown to be presynaptic and require PKA (*Weisskopf et al., 1994*), we tested the possibility that compensation in AKAP7 KO mice might have produced a PKA-independent form of LTP. We pharmacologically inhibited PKA with H89 and observed that the tetanus-induced LTP in AKAP7 KO slices remained dependent on PKA (*Figure 4E*). A previous report found that presynaptic long-term depression (LTD) at MF-CA3 synapses also relies on cAMP/PKA signaling (*Tzounopoulos et al., 1998*). We therefore examined the AKAP7 KO and found that LTD was preserved in the AKAP7 KO mice compared with WT controls (*Figure 4F*). To determine if basal presynaptic function was altered in AKAP7 KO mice, we assessed short-term plasticity while monitoring NMDAR-EPSCs. All measured short-term plasticity dynamics, paired-pulse facilitation (*Figure 4G*) and frequency facilitation (*Figure 4H and I*) were comparable to WT controls. These results are consistent with our anatomical findings that deletion of AKAP7 does not cause gross changes in basal synaptic function. Collectively, our results suggest that cAMP-induced LTP requires high levels of localized PKA signaling in MF axons and presynaptic terminals, whereas tetanus-induced LTP does not.

## Contextual pattern separation is impaired in AKAP7 KO mice

Disruption of DG function in mice has been shown to impair contextual (*McHugh et al., 2007*) and non-cued spatial forms of discrimination (*Gilbert et al., 1998*), commonly referred to as pattern separation. To test for alterations in contextual pattern separation in the AKAP7 KO mice, we first employed a contextual fear discrimination paradigm (*Figure 5A*) (*McHugh et al., 2007*; *Nakashiba et al., 2012*). WT and KO animals received a foot-shock in 'context A' but not in 'context B,' where the two contexts differed only in the flooring and wallpaper. WT mice demonstrated intact contextual pattern separation because they rapidly learned that context B was comparatively safer and reduced their freezing behavior in context B after just one trial while retaining a robust freezing response in context A. However, AKAP7 KO mice took 7–8 trials to distinguish context B from context A (*Figure 5B–D*). The AKAP7 KO mice were also slightly delayed in extinction of freezing behavior (*Figure 5E*), which is thought to involve inhibitory learning (*Myers and Davis, 2007*). Apart from these behavioral differences, KO mice were not impaired in a typical contextual memory task (Barnes maze) and did not display changes in anxiety as assessed by elevated plus maze, open field and marble burying (*Figure 5—figure supplement 1*). There were also no differences in gross motor function as measured by wire hang, pole test, and rotarod (*Figure 5—figure supplement 1*). These results suggest that AKAP7 plays a crucial role in DG-mediated contextual pattern separation without significant effects on other common behavioral paradigms.

Pattern separation behavior has been directly linked to the process of ongoing neurogenesis that occurs in the subgranular zone of the DG leading to a continuous integration of adult-born DGCs into the hippocampal circuit. Inhibition of neurogenesis by irradiation (*Clelland et al., 2009*) or by using genetic approaches (*Nakashiba et al., 2012*; *Scobie et al., 2009*; *Tronel et al., 2012*) results in impaired pattern separation, whereas increasing neurogenesis leads to improved performance in a pattern separation assay (*Creer et al., 2010*; *Sahay et al., 2011*). To test whether neurogenesis could explain the AKAP7 KO behavioral deficiency, we did pulse-chase experiments with 5-bromo-deoxyuridine (BrdU) and quantified BrdU-positive neurons at multiple stages of granule cell differentiation in WT and KO mice. We found no differences in adult neurogenesis either under basal conditions or when enhanced by running wheel access and environmental enrichment (*Figure 5—*

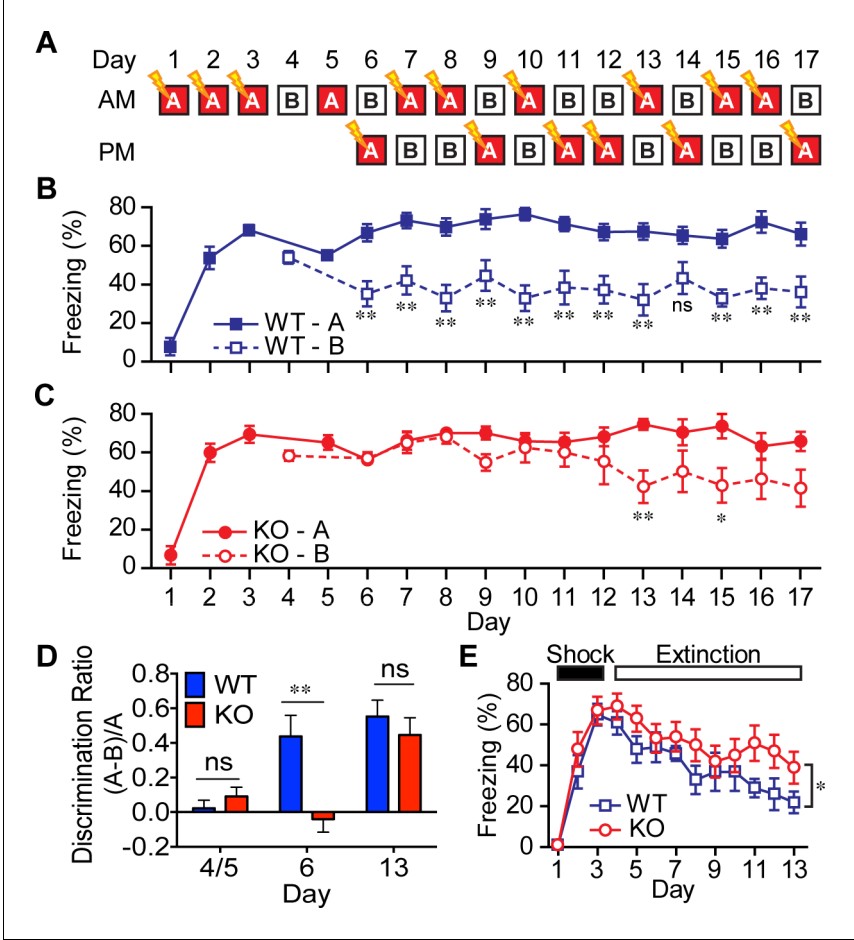

**Figure 5.** AKAP7 KO animals show impaired contextual discrimination. (**A**) Experimental protocol to test contextual fear discrimination. Animals were probed for freezing (% time immobile) in either 'context A' where they received a single footshock (0.65 mA, 2 s) (except Day 5, when there was no footshock) or the no-shock 'context B', which had a different floor and a striped pattern on one wall. (**B**) WT animals exhibited a comparable duration of freezing behavior in both contexts on Days 4 and 5, respectively, but by Day 6 and thereafter demonstrated reduced freezing behavior in 'context B'. (**C**) KO animals did not begin to discriminate between contexts A and B until Day 13. (**D**) Discrimination ratio for Days 5, 6 and 13 demonstrate differences between WT and KO animals on key days (n = 7 animals per group); data are presented as mean ± SEM (p values determined using Student's unpaired t-test in panels B-D). (**E**) Following 3 days of footshock delivery only in 'context A', AKAP7 KO animals demonstrated a slight but significant deficit in contextual fear extinction compared to WT animals (n = 6–8 animals per group). Animals were never introduced to 'context B.' Two-way ANOVA was used for the effect of genotype. *p<0.05, **p<0.01.

The following figure supplements are available for figure 5:

**Figure supplement 1.** AKAP7 KO mice show normal spatial learning, anxiety, and motor behaviors.

**Figure supplement 2.** Adult neurogenesis is normal in AKAP7 KO animals.

---

*figure supplement 2*). It is therefore unlikely that reduced neurogenesis underlies the deficits in contextual pattern separation behavior that we observe in the AKAP7 KO mice.

## Targeted deletion of AKAP7 in DGCs

Results from global AKAP7 KO mice suggested a critical role for AKAP7 in pattern separation, but the effects on behavior could be influenced by AKAP7 in brain regions other than the hippocampus

(*Figure 1—figure supplement 1*). To address this question, we specifically deleted AKAP7 from DGCs using a *Pomc*-Cre driver mouse line (*McHugh et al., 2007*) crossed with *Akap7*[lox/lox] mice (*Jones et al., 2012*). Specificity of Cre recombinase expression was determined by TdTomato reporter (*ROSA26Sor*-lox-STOP-TdTomato x *Pomc*-Cre) (*Figure 6A–B*). TdTomato expression was tightly restricted to DGCs and the arcuate nucleus, consistent with previous reports (*McHugh et al., 2007*; *Sanz et al., 2015*). In order to avoid germ line recombination, we bred *Pomc*-Cre mice onto an *Akap7* KO background and crossed the resulting *Pomc*[Cre/+];*Akap7*[+/-] mice to *Akap7*[lox/lox] mice to generate *Pomc*[Cre/+];*Akap7*[lox/-] (DG-KO) mice. DG-KO animals are heterozygous for *Akap7* in all tissues except those that are Cre-positive, where *Akap7* is expected to be absent. This cross also produced appropriate control mice; i.e., Cre-negative *Akap7*[lox/-] (Het) and Cre-negative *Akap7*[+/lox] (WT). Immunofluorescence microscopy showed delocalization of PKA-RIIβ and loss of AKAP7 in the DGCs of DG-KO animals, whereas localization and expression in Het animals were normal (*Figure 6C*). Expression of AKAP7 in the thalamus (*Figure 6C*) and other brain regions was unaffected in the DG-KO. These results further demonstrate that the AKAP7 detected in the DG molecular layer in WT mice arises from DGCs and not axonal projections of the perforant path.

We examined whether DG-KO animals would recapitulate the behavioral and electrophysiological deficits seen in the global AKAP7 KO. MF-CA3 synaptic transmission was monitored using extracellular field recordings in acute hippocampal slices. The DG-KO mice exhibited an impairment in cAMP-induced LTP, although the decrease was not as complete as in the global AKAP7 KO (*Figure 6D*). This likely results from some residual levels of AKAP7 and RIIβ present in the MF pathway due to incomplete Cre recombination. As in the global KO, the synaptically-induced LTP remained intact in the DG-KO (*Figure 6E*). These results confirm that presynaptically localized AKAP7 and its anchored PKA-RIIβ are essential for cAMP-induced LTP.

## Non-cued spatial pattern separation is impaired in DG-KO mice to the same extent as the global KO

To determine whether AKAP7 deletion specifically in DGCs impairs pattern separation, we utilized another test of DG function, delayed non-matching-to-place (DNMP). This assay determines the ability of mice to recognize spatial differences in the location of a food reward within a radial arm maze without relying on visual cues. This test involves reward-based learning rather than the contextual fear discrimination paradigm employed in *Figure 5* and has also been used extensively to study the role of DGCs in pattern separation (*Clelland et al., 2009*; *Guo et al., 2011*, *2012*; *Nakashiba et al., 2012*). We compared all four genotypes: WT, global KO, DG-KO and Het. Animals were first trained to find a food reward (R) at the end of one open arm after being placed in the start (S) arm. After finding the reward, mice were removed and a third arm was opened and baited. The mice learned to ignore the previously baited arm and go to the third open arm for reward. Mice were then returned to the maze to test whether they could correctly identify this new reward arm while ignoring the previous reward arm location (*Figure 7A*). For this test, arms that are closer together (2-arm) are more difficult to discriminate between compared to arms that are further apart (4-arm) and the 2-arm test robustly detects DGC defects (*Clelland et al., 2009*). AKAP7 KO and DG-KO animals committed significantly more errors in the 2-arm configuration compared to WT and Het over the 5-day experiment (*Figure 7B*). In the 4-arm separation task, AKAP7 KO and DG-KO animals performed similar to WT and Het animals during the fourth and fifth day of training. Critically, the specific loss of AKAP7 from DGCs was sufficient to disrupt non-cued spatial pattern separation to the same degree as observed for the global KO when animals were challenged in the 2-arm test (*Figure 7C*). These results demonstrate that defective pattern separation is directly linked to AKAP7 function in DGCs.

## Discussion

Our study has identified the first presynaptic AKAP involved in anchoring PKA in axonal projections in adult neurons. AKAP7α localizes PKA-RIIβ presynaptically in the axons and terminals of MF projections to the CA3 pyramidal cells of the hippocampus. We used genetic disruption of AKAP7 first globally and then specifically in DGCs to explore the function of this presynaptic AKAP and the colocalized PKA. We find that loss of AKAP7 impairs the localization of PKA in the MF projections and leads to a defect in pattern separation behaviors that have been shown to depend specifically on

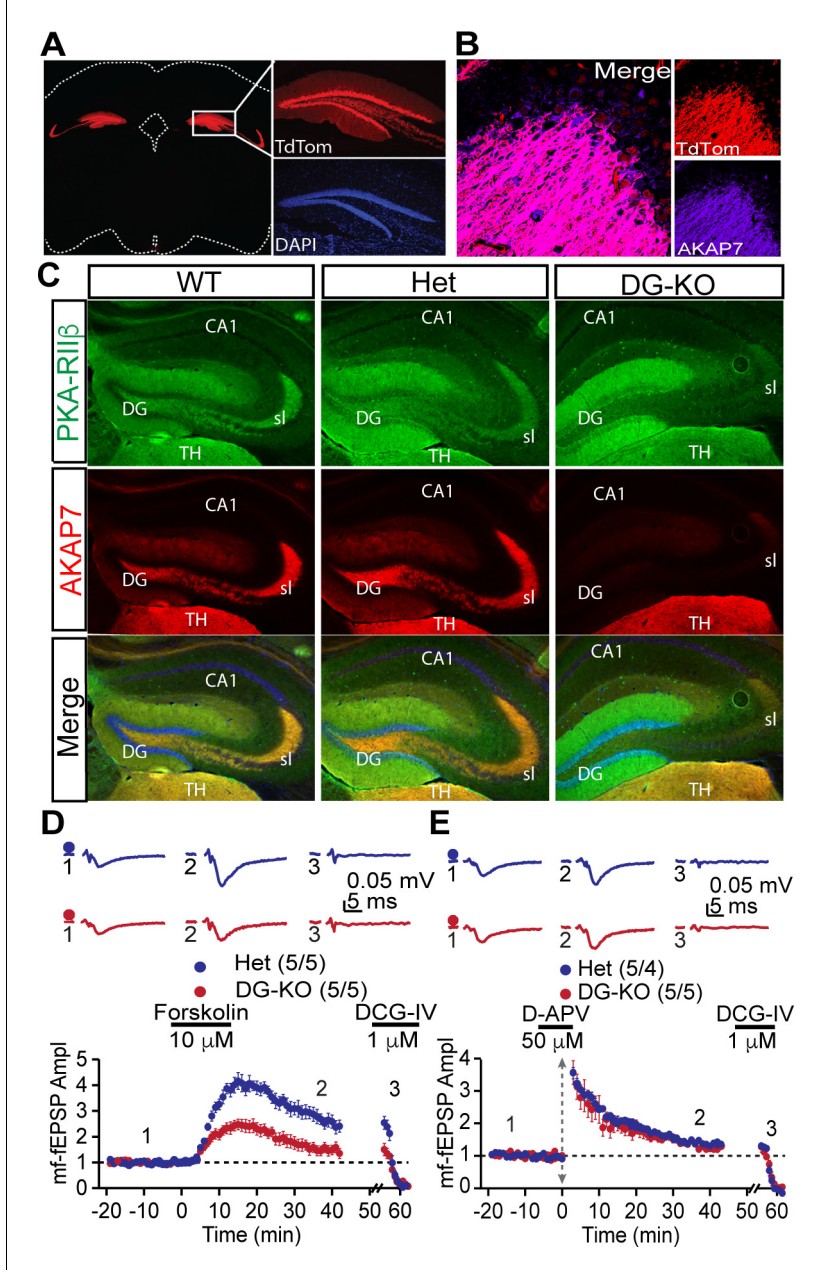

**Figure 6.** DG-specific AKAP7 KO shows plasticity deficits similar to global KO. (**A**) Fluorescent protein TdTomato expression in the hippocampus of *Pomc*[+/Cre] crossed to TdTomato reporter mice is restricted to dentate granule neurons. (**B**) Merged image of TdTomato expression and AKAP7 staining show that *Pomc*-Cre was active in the majority of AKAP7-expressing DGCs. (**C**) *Pomc*[+/Cre];*Akap7*[lox/-] (DG-KO) animals lose AKAP7 in the dentate gyrus but not the neighboring thalamus (TH) (or other brain regions; not shown). PKA-RIIβ is displaced from MFs in DG-KO animals, although some staining is still present in stratum lucidum likely due to incomplete Cre recombination. *Akap7*[lox/-] (Het) animals show AKAP7 expression and PKA-RIIβ distribution comparable to WT. (**D**) cAMP-induced MF-LTP is impaired in DG-KO animals, similar to global AKAP7 KO. Het (control): 2.73 ± 0.25 of baseline vs. DG-KO: 1.54 ± 0.20 of baseline; p=0.0053 (Student's unpaired t-test); Cohen's d = 2.40. Experimental conditions are as in *Figure 4B*. (**E**) Tetanus-induced MF-LTP remains intact in DG-KO compared with Het control. We also assessed basal synaptic function by measuring the paired pulse ratio (with a 40 ms interspike interval) in the DG-specific KO and found no significant difference from the control (control HET: 2.34 ± 0.08 vs DG-KO: 2.3 ± 0.09, p=0.76, t-test, n = 10. Experimental conditions are as in *Figure 4*. For each group of mice, the values in parentheses correspond to the number of slices/number of mice.

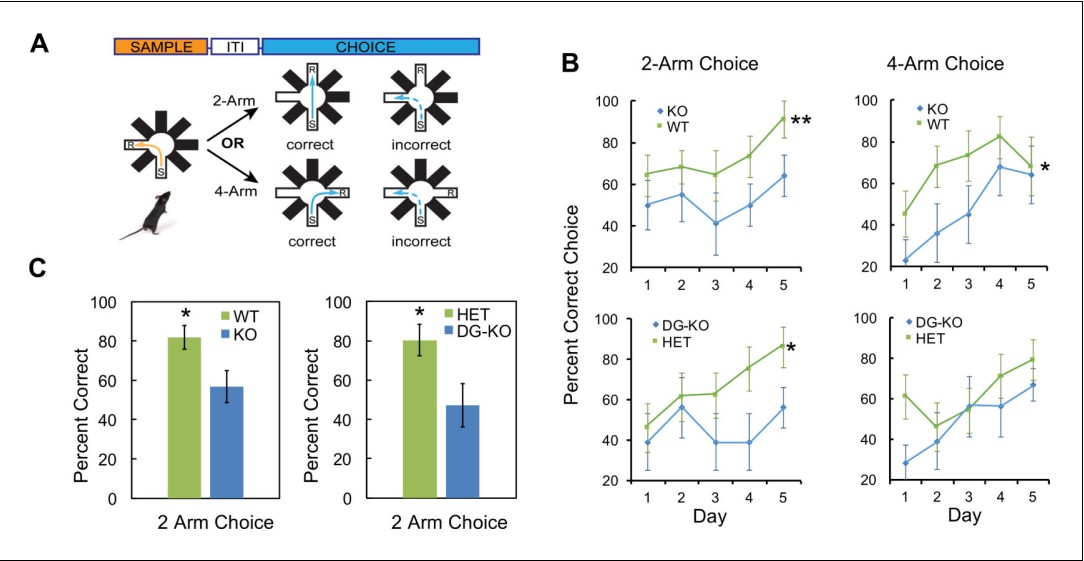

**Figure 7.** AKAP7 KO and DG-KO animals demonstrate impaired non-cued spatial pattern separation. (**A**) Experimental protocol to test non-cued spatial discrimination by delayed non-matching-to-place (DNMP) using an 8-arm radial arm maze. Food restricted animals were placed in the start (S) arm and the froot loop reward (R) placed in the goal arm. (**B**) Learning curves for each genotype over a 5-day experiment. Animals were randomly assigned two trials in the two-arm separation and two trials in the four-arm separation each day and the results averaged and expressed as percent correct choice. The results were analyzed by two-way ANOVA and showed an effect of genotype in the two-arm separation (KO:WT, p=0.004, DG-KO:HET, p=0.013). In the four-arm separation the KO:WT comparison was significant, (p=0.012); the DG-KO:HET comparison was not significant (p=0.095). (**C**) AKAP7 global KO and DG-KO animals discriminate poorly compared to their respective controls in the more difficult two-arm separation test. The performance on days 4–5 in the two-arm separation test was pooled and analyzed by two-tailed, unpaired T-test (WT:KO, p=0.023, Cohen's d = 1.04; HET:DG-KO, p=0.025, Cohen's d = 1.01). In the four-arm separation trials the WT:KO and HET:DG-KO comparisons did not show significant differences (p=0.565 and p=0.346 respectively). WT and HET groups in both the two-arm and four-arm trials performed significantly better than 50% at days 4–5 (p<0.05), whereas the KO and DG-KO did not. KO (n = 11), WT (n = 11), HET (n = 14), and DG-KO (n = 9). *p<0.05, **p<0.01.

DGCs (*McHugh et al., 2007*). In addition, we find that the AKAP7 KO displays a dramatic loss of cAMP-induced LTP at the MF-CA3 synapse.

There are multiple circuits in which presynaptic PKA has been shown to play an important functional role in synaptic plasticity including the hippocampal MF-CA3 synapses (*Nicoll and Schmitz, 2005*), the hippocampal CA3-CA1 synapse (*Park et al., 2014*), cortico-thalamic synapses (*Castro-Alamancos and Calcagnotto, 1999*), corticostriatal synapses (*Spencer and Murphy, 2002*) and the parallel fiber synapses in the cerebellum (*Salin et al., 1996*). In the hippocampus, AKAP7α is primarily restricted to DGCs, although it is expressed in other brain regions including the cortex, striatum, thalamus and cerebellum (*Figure 1—figure supplement 1*). In MF axons, AKAP7 is apparently the only anchor for PKA-RIIβ since the two proteins colocalize throughout the axons and terminals and the loss of AKAP7 results in delocalization of PKA-RIIβ and C subunits from these presynaptic projections as indicated by a strong reduction in immunofluorescence. In contrast, dendritic and cytoplasmic PKA-RIIβ increases in the AKAP7 KO. Dendritic PKA remains localized in the AKAP7 KO and is likely to bind other postsynaptic AKAPs including AKAP5 (*Sanderson and Dell'Acqua, 2011*; *Tunquist et al., 2008*; *Weisenhaus et al., 2010*), AKAP12 (gravin) (*Havekes et al., 2012*) and MAP2 (*Zhong et al., 2009*). We do not know whether AKAP7 plays any specific role in dendrites that cannot be compensated for by these other AKAPs. The modest elevation in dendritic and somatic PKA in the AKAP7 KO could potentially compete for cAMP during postsynaptic signaling in response to perforant path stimulation from the entorhinal cortex, but this seems unlikely given the highly organized domain structure of PKA signaling complexes in the postsynaptic density.

Our understanding of the role of the DG in memory and learning has developed significantly over the past 15 years (*Kesner and Rolls, 2015*). Early studies found that colchicine induced lesions in the DG, but not CA1 regions of the hippocampus in rats, produced deficits in spatial discrimination (*Gilbert et al., 2001*). Genetic disruption of the NR1 subunit of the NMDA receptor specifically in DGCs results in a defect in discrimination between similar contexts (pattern separation) in mice (*McHugh et al., 2007*). The discovery that neurogenesis is occurring in the subgranular zone of the dentate gyrus and adult-born DGCs are constantly integrating into the DG circuits (*Altman and Das, 1965*; *van Praag et al., 2002*) led to intensive efforts to understand the function of the DG and determine whether these adult born DGCs play a distinct role in contextual discrimination (*Aimone et al., 2011*; *Sahay et al., 2011*). Although the mechanisms are not resolved, the studies indicate that adult born DGCs are preferentially involved in accurate pattern separation, and the mature DGCs are required for pattern completion or generalization (*Danielson et al., 2016*; *Nakashiba et al., 2012*). We found that deleting AKAP7 did not affect adult neurogenesis, but the function of AKAP7 in pattern separation could preferentially involve these adult born DGCs. In summary, our study identifies an important functional role for AKAP-dependent subcellular localization of PKA in DGCs.

MF-LTP can be induced by repeated trains of tetanic stimulation or direct activation of cyclases with forskolin to elevate cAMP. Early efforts to understand the role of cAMP and its effectors in MF-LTP relied on pharmacological and genetic tools that affected cAMP and cAMP effectors throughout the DGCs. Bath application of PKA inhibitors during tetanus blocks the induction of MF-LTP (*Huang et al., 1994*; *Weisskopf et al., 1994*). The role of adenylyl cyclases was demonstrated by genetic deletion of the calcium-stimulated cyclases, AC8 and AC1. Deletion of AC1 or AC8 reduced tetanus-induced MF-LTP and the AC1/AC8 double KO completely eliminated tetanus induced LTP. It is interesting to note that the LTP induced by direct cyclase activation with forskolin was unaffected in the AC1/AC8 double KO (*Wong et al., 1999*) suggesting that forskolin is acting on another cyclase in the DGCs to induce LTP. Targeted disruption of either the Cβ or RIβ subunit gene of PKA inhibited tetanus-induced LTP at MF-CA3 synapses without decreasing cAMP-induced LTP (*Huang et al., 1995*). Recently, the contribution of another cAMP effector, Epac2, was examined. Epac2 is a cAMP-regulated guanine nucleotide exchange factor (GEF) for Rap1/2, and electrophysiological recordings from Epac2 KO mice demonstrate defects in both tetanus- and cAMP-induced MF-LTP (*Fernandes et al., 2015*). These approaches clearly establish that cyclases, PKA, and Epacs are involved in the mechanisms that underlie MF-LTP, but do not provide an understanding of the importance of subcellular localization of signaling complexes in axons, soma or dendrites of DGCs. Our study demonstrates that AKAP7 anchors PKA in the axonal projections of DGCs and coordinates the cAMP-induced LTP pathway without affecting the PKA dependent, synaptically-induced MF-LTP. This surprising result suggests that localization of PKA could play a differential role in these two different modes of eliciting LTP. One possibility is that synaptically induced LTP is more dependent on the axonal transport of synaptic vesicle proteins that might be phosphorylated in the soma and delivered to the terminals. Rates for transport of these Golgi-derived vesicles are as high as 2–5 μm/s (*Brown, 2000*), and they could therefore travel the ~1000 μm distance from the DGC body to the terminals and play a role in the sustained LTP seen 30–40 min after tetanic stimulation. Severing the MF axons near the DGC body prevents synaptically induced LTP without affecting cAMP-induced LTP further supporting a role for the cell body in maintenance of synaptically induced LTP (*Calixto et al., 2003*). RI and RII isoforms of PKA have different anchoring characteristics (*Di Benedetto et al., 2008*) with PKA-RI typically concentrated in the soma of neurons (*Yang et al., 2014*), while PKA-RIIβ is usually tightly anchored to AKAPs in dendrites and, in this study, axonal projections. This leads us to speculate that tetanus-induced LTP depends on PKA activation in the somatodendritic compartment and cAMP-induced LTP depends on PKA-RIIβ in the MF projections. The expression of AKAP7 in other brain regions suggests a possible PKA anchoring role in other presynaptic pathways where PKA-dependent LTP has been observed and our *Akap7*^lox/lox mouse line will facilitate these future studies.

Experience-induced changes in synaptic plasticity (LTP and LTD) are the most attractive cellular substrates to explain learning and memory, but it has been exceedingly difficult to make causal connections between synaptic plasticity and behavior. The function of the LTP mechanism that can be directly engaged by forskolin and is disrupted in the AKAP7 KO is unknown. One possibility is that direct activation of localized Gs-coupled receptors, cyclases and PKA in the axons and presynaptic

terminals could alter the excitability of those MF projections and might act synergistically with perforant path induced DG excitation. Extrinsic neuromodulatory projections from the locus coeruleus terminate in the stratum lucidum (*Amaral et al., 2007*) and may provide noradrenergic modulation of MF-LTP (*Huang and Kandel, 1996*). There is evidence for synergism between the β-adrenergic receptor agonist, isoproterenol, and repetitive synaptic stimulation of MF-LTP in slice culture (*Huang and Kandel, 1996*). More recently, it has been observed that a weak tetanic stimulation of the MF-CA3 pathway *in vivo* can synergize with lateral ventricle administration of isoproterenol to elicit LTP lasting more than 24 hr (*Hagena and Manahan-Vaughan, 2012*). In addition, the pituitary adenylate cyclase activating peptide (PACAP) receptor is highly expressed in DGCs and this Gs-coupled GPCR is localized to MF projections (*Otto et al., 1999*). Targeted disruption of the PACAP receptor (*Adcyap1r1*) in neocortex and hippocampus led to deficits in both MF-LTP and contextual fear conditioning (*Otto et al., 2001*). We propose that AKAP7-localized PKA provides an intracellular signaling mechanism to potentiate the actions of neuromodulators targeting G-protein-coupled receptors expressed specifically on the MF projections. DG dysfunction has been implicated in a wide range of psychiatric disorders including post-traumatic stress disorder, depression and addictive behavior (*García-Fuster et al., 2013*; *Kheirbek et al., 2012*). Our studies suggest that alterations in presynaptically anchored PKA can disrupt specific aspects of DG function and could play a role in these disorders.

## Materials and methods

### Generation of Akap7 mutant mice

Generation of AKAP7 KO and floxed AKAP7 mouse lines was described previously (*Jones et al., 2012*). We generated tissue-specific DG-KO animals by crossing *Akap7*^lox/lox to *Pomc*-Cre^+;*Akap7*^+/- animals, which express Cre recombinase primarily in the hypothalamus and dentate gyrus. This cross generates four genotypes: *Pomc*-Cre^+;*Akap7*^lox/- (global Het, DG-KO), Cre^-;*Akap7*^lox/- (global Het), Cre;*Akap7*^+/lox (WT), and *Pomc*-Cre^+;*Akap7*^+/lox (DG-Het; not used). Mice were given access to food pellets and water ad libitum, and maintained on a 12:12 hr light/dark cycle. All procedures were approved by the Institutional Animal Care and Use Committee of the University of Washington and conformed to NIH guidelines. For behavioral and electrophysiological experiments, the experimenter was blind to genotype and un-blinded after completing data analysis. Both male and female mice were used in experiments, and no significant differences were seen between sexes in behavioral or electrophysiological studies so the N values include both sexes.

### Immunoblotting

Hippocampi were rapidly dissected and homogenized for immunoblotting following previously described methods (*Jones et al., 2012*). Antibodies used were as follows: rabbit anti-AKAP7 (Proteintech, Chicago, IL), mouse anti-PKA-RIIβ (BD Biosciences, San Jose, CA) and mouse anti-SP1 (07–645, Millipore, Billerica, MA).

### Immunohistochemistry and imaging

Brains were fixed and processed for standard immunohistochemistry. Neuron subregions were identified using antibodies specific for each: dendrites, anti-MAP2 (Thermo Scientific, Waltham, MA); axons, anti-Neurofilament-M (Developmental Studies Hybridoma Bank, Iowa City, IO); presynaptic boutons, anti-Znt3 (Synaptic Systems, Goettigen, Germany) and anti-synaptophysin (Millipore); postsynaptic densities, anti-PSD-95 (Alomone, Jerusalem, Israel). Citrate buffer antigen retrieval was required to recover AKAP7 (Proteintech) immunostaining. PKA subunits were visualized using anti-PKA-RIIβ (BD Biosciences) and anti-PKA-Cα (gift from the laboratory of Susan Taylor). Fluorescent secondary antibodies were utilized (Molecular Probes, Eugene OR). Coverslips were mounted with ProLong Gold (Molecular Probes) or Fluoromount G (Electron Microscopy Sciences, Hatfield, PA) with or without DAPI. Slides were imaged in the University of Washington Keck Microscopy Center on a Nikon Eclipse E600, Zeiss 510 META, or Leica SL confocal microscopes. For quantification of colocalization with AKAP7, regions of interest were chosen from the stratum lucidum of images taken at high magnification by confocal microscopy (Leica SL Confocal), n = 4–11 per synaptic marker. Each set was analyzed using a colocalization plugin Coloc two in Fiji-ImageJ

(*Schindelin et al., 2012*). The correlation between the red and green images is given as intensity correlation quotient (ICQ) values. ICQ values are distributed between −0.5 and +0.5 such that random localization gives an ICQ of approximately 0. Colocalization gives positive ICQ values and segregated localization gives negative ICQ values.

## Contextual fear discrimination and extinction

Assay was performed using a paradigm similar to previous reports (*McHugh et al., 2007*; *Nakashiba et al., 2012*). Briefly, mice were placed in a shock chamber (Colbourn Instruments, Whitehall, PA) placed inside a fume hood to provide background noise. Animals were acclimated to handling and transportation prior to testing. Trials consisted of two types: context A (shock) and context B (non-shock). The two chambers differed only in the floor (white plastic-B versus shock grid-A) and wallpaper (striped wallpaper-B versus white-A). For context A, animals were placed in the chamber and allowed to explore for 3 min, given a 2 s, 0.65 mA foot shock and removed 1 min after foot shock. Context B followed the same procedure, except that animals received no foot shock. Testing in A occurred once daily for 3 days. Between every trial, the chamber was cleaned with 5% NaOH and the floor scented with 1% Acetic Acid. On day 4, animals experienced only context B. On day 5, animals experienced context A without a foot shock. On the remaining days 6–17, animals were placed in each context once daily, presented in pseudorandom order. Animals were always shocked at the end of their time in context A but never in context B. Video recordings were analyzed by Noldus EthoVision software. Contextual fear extinction was carried out similarly, except that animals only received foot shocks on days 1–3 and were never introduced to a second context.

## Delayed non-matching-to-place radial eight-arm maze (DNMP-RAM)

We used a behavioral assay procedure similar to one previously described (*Clelland et al., 2009*). Briefly, mice were singly housed and food restricted to 85–90% of initial body weight. Testing included four trials per day over 5 consecutive days. Each trial included two phases, a sample phase and a choice phase. For the sample phase, only the sample and start arms were open, separated by 90-degrees, with a food reward placed at the end of the sample arm (Froot Loops, Kellogg's, Battle Creek, MI). Mice were placed in the maze facing the end of the start arm and allowed 3 min to find the food reward, after which they were removed and returned to their home cage. The maze was then rotated 90 degrees and cleaned with 1% acetic acid. New start and sample arms were opened in the same relative locations as during the sample phase. A third arm, the baited choice arm, was also opened, separated from the sample arm by 1 (2-arm separation) or 3 (4-arm separation) closed arms. 2- and 4-arm separations were presented in random sequence each day. An observer blinded to genotypes scored correct or incorrect choices. Data were analyzed by two-way ANOVA or t-test as described in the figure legends.

## Accelerating rotarod

Mice were placed on a rotarod apparatus (Ugo Basile, Varese, Italy) and its speed of rotation increased from 3–40 rpm over 5 min. Mice experienced 5 trials/day over 3 days, with a 20 min intertrial interval. The apparatus automatically recorded the latency to fall. N = 8 for each genotype.

## Wire hang

Mice were placed on an upside-down wire bar cage lid (identical to that used for standard housing), the lid shaken three times to encourage the mouse to grip the wire, the lid turned right-side up so that the mouse hung underneath, and the lid positioned 50 cm above a cage filled with 5 cm of bedding. We measured the latency to fall and terminated the first two trials after 60 s if the mouse had not already fallen, but allowed the third trial to continue for up to 10 min. Trials were repeated 20 min apart. If an animal fell before 10 s in any trial, then the trial was restarted until the animal either hung for at least 10 s or failed three consecutive attempts. We weighed mice after testing and calculated the impulse (g x s) as animal weight (g) x latency to fall (s). N = 13 for each.

## Pole test

Mice were positioned nose up near the top of a 57.5 cm tall, 1.3 cm diameter steel pole wrapped with surgical tape to provide some grip. The pole was positioned in the center of a 26 (W) X 47.6 (L)

X 30 (H)-cm cage filled with 6 cm of bedding. We measured the latency to turn around and face downward and the time to descend. N = 9 wt, 11 KO.

## Open field

Mice were placed in a 80 cm diameter arena in bright room light and allowed to explore for 4 min. Video recordings were analyzed by Noldus EthoVision to measure the percent of time spent in and the number of entries into the center area. N = 13 WT and 15 KO.

## Marble burying

Mice were placed in a 15 (W) X 27 (L) X 12 (H)-cm cage with 15 evenly spaced, 1.4 cm diameter, blue marbles atop 5 cm of bedding. We counted the number of marbles completely buried after 20 min. N = 13 for each.

## Elevated plus maze

To test basal anxiety, we used an elevated plus maze consisting of opposing arms either open (without walls) or closed (with walls) and elevated 75 cm from the ground. Animals were habituated animals to handling for 5 min each day for 3 days prior to the experiment. For testing, animals were placed in the center of the maze facing one of the open arms and allowed to explore freely for 8 min. Time spent in open arms was measured using Noldus EthoVision software of the acquired video recording. The experiment was carried out under low light conditions ( ~20 lux). N = 12 for each.

## Barnes maze

To assess cued spatial learning, we use a modified Barnes maze consisting of a disc 1 m in diameter raised 75 cm off the ground with 18 equivalent holes along its perimeter, one of which leads to a contained escape with food pellet. For testing, the animal is initially placed in the center of the maze and allowed to explore for 5 min, three times per day, over 4 days, with a 2-min intertrial interval (ITI) followed by three probe trials on the fifth day. Motivation to escape is provided by bright lighting. Animals unable to locate the escape within the allotted 5 min during training were placed in the escape for 1 min. Latency to escape, time spent in target quadrant, and acquisition were all determined by Noldus Ethovision analysis of acquired video recording.

## Electrophysiology

Experiments were carried out as in previous work (*Kaeser-Woo et al., 2013*). Briefly, mice were deeply anesthetized with isoflurane, decapitated, and acute, transverse hippocampal slices (400 μm) were prepared with a VT1200S microtome (Leica). Experiments were carried out at 25 ± 0.5°C. To monitor synaptic transmission and plasticity, conventional field and whole-cell recordings were made with a Multi-Clamp 700B amplifier (Molecular Devices). Stimulation and acquisition were controlled by custom software. Output signals from recordings were acquired at 5 kHz, filtered at 2.4 kHz, and stored in IgorPro (Wavemetrics). Summary data are expressed as mean ±SEM. Statistics were performed using OriginPro (Origin Lab), and statistical significance was determined by the Student's t test. Only statistically significant differences are indicated. To elicit long-term plasticity, two trains containing 125 pulses at 25 Hz, 20 s inter-train interval, were delivered via the stimulating pipette. Alternatively, 10 μM forskolin, an activator of adenylyl cyclase, was bath applied for 10 min and then washed out. The magnitude of long-term plasticity was determined by comparing baseline-averaged responses before induction with the last 10 min of the experiment. Example traces are averages of at least 30 consecutive sweeps taken from a single representative experiment.

## RiboTag isolation and analysis of mRNA transcripts

RiboTag experiments were performed as described (*Sanz et al., 2009*). Briefly, mice were bred to selectively express a floxed HA-tagged ribosomal protein, RPL22, by crossing the RiboTag mouse with a dentate gyrus-specific Cre driver mouse, *Pomc*-Cre (*McHugh et al., 2007*). To isolate DGC-specific transcripts, mice were sacrificed and the hippocampi were rapidly dissected and flash frozen. Individual tissue samples were dounce homogenized in a buffer supplemented with RNase inhibitors. Polysomes were immunoprecipitated using antibodies to HA (Covance) and protein A/G-coupled magnetic beads (Pierce BioTechnologies). Input and immunoprecipitated (IP) RNAs were quantified

by Quant-It RiboGreen (Life Technologies). Quantitative reverse transcriptase-PCR was performed using SYBR green One-Step (Agilent). Oligonucleotides selected using the PrimerBank database: Desmoplakin (*Dsp*): For: 5'-GGATTCTTCTAGGGGAGACTCAGT-3', Rev: 5'-CCACTCGTATTCCGTC TGGG-3', Proximal 1 (*Prox1*): For: 5'-AGAAGGGTTGACATTGGAGTGA-3', Rev: 5'-TGCGTG TTGCACCACAGAATA-3', Glutaminase (*Gls*): For: 5'-CTACAGGATTGCGAACATCTGAT-3', Rev: 5'- ACACCATCTGACGTTGTCTGA-3', Phosphodiesterase 1a (*Pde1a*): For: 5'-CCGGGATTGGTTGGC TTCAA-3', Rev: 5'- AATGCTGCGAAACTTTGGTTTT-3', CNPase (*Cnp*): For: 5'-TTTACCCG-CAAAAGCCACACA-3', Rev: 5'- CACCGTGTCCTCATCTTGAAG-3', Transferrin (*Trf*): For: 5'-GCTG TCCCTGACAAAACGGT-3', Rev: 5'-CGGAAGGACGGTCTTCATGTG-3'. β-actin (*Actb*): For: 5'-GGC TGTATTCCCCTCCATCG-3', Rev: 5'-CCAGTTGGTAACAATGCCATGT-3'. *Akap7* isoform-specific primers as described previously (*Jones et al., 2012*).

## Data analysis and statistics

No statistical methods were used to determine sample sizes prior to experiments. Multiple measurements from the same mouse in the same experiment were considered technical repeats and were averaged. This average was considered a single biological repeat and was used to determine sample size for statistical analysis.

## Acknowledgements

We thank Drs. Linghai Yang and Jonathan Bean for their helpful comments on this work and thank Larry Zweifel, and Richard Palmiter for sharing behavioral equipment. JD was supported by NIGMS Pharmaceutical Sciences Training Grant 5-T32-GM007750 at the University of Washington.

## Additional information

### Funding

| Funder | Grant reference number | Author |
|---|---|---|
| National Institute of General Medical Sciences | 5-T32-GM007750 | Jennifer Deem |
| National Institutes of Health | R01-DA017392 | Pablo E Castillo |
| National Institutes of Health | R01-MH081935 | Pablo E Castillo |
| National Institutes of Health | R01-GM032875 | G Stanley McKnight |

The funders had no role in study design, data collection and interpretation, or the decision to submit the work for publication.

### Author contributions

BWJ, JD, TJY, MW, Conception and design, Acquisition of data, Analysis and interpretation of data, Drafting or revising the article; CAS, MCS, JC, DN, AM, Acquisition of data, Analysis and interpretation of data; PEC, GSM, Conception and design, Analysis and interpretation of data, Drafting or revising the article

### Author ORCIDs

G Stanley McKnight, http://orcid.org/0000-0001-9531-2936

### Ethics

Animal experimentation: This study was performed in strict accordance with the recommendations in the Guide for the Care and Use of Laboratory Animals of the National Institutes of Health. All of the animals were handled according to approved institutional animal care and use committee (IACUC) protocol (2022-01) of the University of Washington.

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
