## [Decision Letter]

Thank you for submitting your article "Targeted deletion of AKAP7 in dentate granule cells impairs spatial discrimination" for consideration by *eLife*. Your article has been reviewed by three peer reviewers, one of whom, Mary B Kennedy (Reviewer #1), is a member of our Board of Reviewing Editors and the evaluation has been overseen by Gary Westbrook as the Senior Editor. The following individuals involved in review of your submission have agreed to reveal their identity: Mark Dell'Acqua (Reviewer #3).

The reviewers have discussed the reviews with one another and the Reviewing Editor has drafted this decision to help you prepare a revised submission.

Summary:

The manuscript shows convincingly that AKAP7, specifically, the α form of AKAP7, is expressed in the axons and mossy fiber terminals of dentate granule cells in the hippocampus, and that deletion of the AKAP7 gene leads to a dramatic loss of localization of the RIIβ form of PKA in the axons and terminals of DGCs located in stratum lacunosum. They go on to show that both global deletion of AKAP7 and a targeted deletion of AKAP7 in DGCs alone interferes with a specific behavior that has been shown to require the mossy fiber pathway, contextual pattern separation. Thus, the function of AKAP7, which is to localize RIIβ-PKA to DGC axons and terminals, is necessary for animals to display proper contextual pattern discrimination. This appears to be the first time that localization of a particular PKA isoform within neurons by an AKAP has been shown to influence a high level behavior. The study is elegantly done and very interesting. However, the reviewers have several suggested revisions that would improve the manuscript.

Essential revisions:

1) The authors do not discuss earlier evidence about a mechanistic distinction between forskolin-induced and tetanus-induced LTP at the mossy fiber terminals until late in the discussion, leaving one reviewer temporarily puzzled about the data in Figure 4 that shows a difference in the effect of AKAP7 deletion on LTP induced in these two ways. It would be preferable to move the discussion of the apparently different mechanisms underlying these two forms of presynaptic plasticity to the introduction, or at least to an introductory paragraph motivating the experiments shown in Figure 4. One important contribution of the study is the requirement for AKAP7 in direct cAMP induction of PKA-dependent MF-LTP, but not tetanus-induced MF-LTP (which is also PKA-dependent). The Discussion section makes this clear, and the work provides the impetus for future studies characterizing the different roles of AKAP-PKA signaling in DGC cell bodies versus MF terminals in plasticity regulation as well as roles for neuromodulators acting on local PKA signaling in axon terminals. However, for more general readers, this aspect of the study should be explained more explicitly in the Introduction, and results.

2) In the consultation following the reviews, one reviewer raised the point that the dramatic increase in PKA-RIIβ in the dendrites of the granule cells might act to inhibit PKA in these regions, such that the physiological increase in cAMP in the dendrites is insufficient to fully activate the enzyme. This might disrupt LTP induction, particularly late-LTP induction, in the perforant path-DG synapse. For example, four cAMP molecules are required to bind simultaneously to the dimer of R subunits to release active catalytic units, the increase in R2C2 complexes in the somas would have the effect of suppressing activation of PKA because the bound cAMP would be spread over more tetramers, and it would take more total cAMP to activate the same number of catalytic units. However, another reviewer pointed out that since most type II PKA signaling takes place in micro/nanodomains anchored to AKAPs, where cAMP concentrations are locally controlled by AC and PDE activities typically found in these same complexes, as opposed to free in the bulk cytoplasm, the extra Type II PKA in the DG neuron soma/dendrites of AKAP7KO mice is not likely to inhibit normal regulation of AKAP-anchored Type II PKA signaling in those locations. With AKAP anchoring sites likely already being fully occupied, the extra pool of Type II PKA will most likely be cytosolic and primarily found outside these AKAP-organized micro/nanodomains. Signaling by Type I PKA, either unanchored in the cytoplasm (where RI is usually more abundant than RII) or anchored to RI preferring AKAPs, is also not likely to be inhibited by competition for cAMP binding from extra cytosolic type II PKA, both because RI has a higher cAMP affinity than RII and RII is not likely to displace RI from the few known RI preferring AKAPs, which bind RI with greater affinity than RII. This is in contrast to the majority of other AKAPs that are RII preferring, like AKAP7. If the authors agree with this reasoning, the issue should be addressed in the discussion, along with the possibility that a change in plasticity at the DGC dendritic or somal synapses could contribute to the behavioral effect (Although this is less likely than the direct explanation.).

3) For the two Figure 4, and for the behavioral studies, it would be worthwhile to calculate a measure of effect size, such as Cohen's d, for differences between WT and KO's. The effects are significant, but small. Thus, a statement of Cohen's d, which should be larger than about 0.4, would give a more complete statistical picture of the data. In addition, the Figure 7 comparisons should be against 50% chance to make sure the "successful" group tests are indeed better than 50%. Group comparisons are not enough. Also, the authors should add, in both Figure 5 and Figure 7, a genotype X test ANOVA comparison (e.g., testing% correct if the authors gave more than one test per mouse in Figure 7).

4) Although in Figure 2 it certainly appears that AKAP7 staining in WT mice overlaps more with MF axonal and presynaptic markers (NF-M; ZNT3; SYP) than dendritic and postsynaptic markers (MAP2; PSD-95), adding some quantification of relative AKAP7 co-localization with these markers is needed to support these claims. Likewise for Figure 3, some co-localization quantification should be provided to support the authors' statement that AKAP7 KO results in substantial loss of PKA-Calpha co-localization with NF-M.

---

## [Author Response]

*Essential revisions:*

*1) The authors do not discuss earlier evidence about a mechanistic distinction between forskolin-induced and tetanus-induced LTP at the mossy fiber terminals until late in the discussion, leaving one reviewer temporarily puzzled about the data in Figure 4 that shows a difference in the effect of AKAP7 deletion on LTP induced in these two ways. It would be preferable to move the discussion of the apparently different mechanisms underlying these two forms of presynaptic plasticity to the introduction, or at least to an introductory paragraph motivating the experiments shown in Figure 4. One important contribution of the study is the requirement for AKAP7 in direct cAMP induction of PKA-dependent MF-LTP, but not tetanus-induced MF-LTP (which is also PKA-dependent). The Discussion section makes this clear, and the work provides the impetus for future studies characterizing the different roles of AKAP-PKA signaling in DGC cell bodies versus MF terminals in plasticity regulation as well as roles for neuromodulators acting on local PKA signaling in axon terminals. However, for more general readers, this aspect of the study should be explained more explicitly in the Introduction, and results.*

We have briefly mentioned the distinction between forskolin-induced and tetanus-induced LTP at the MF terminals in the Introduction and referenced the differential effects of specific cyclase and PKA mutations. The finding that the dependence on PKA localization for the two different forms of LTP was different was a surprise to us and not anticipated from previous work and this is addressed in the Discussion.

*2) In the consultation following the reviews, one reviewer raised the point that the dramatic increase in PKA-RIIβ in the dendrites of the granule cells might act to inhibit PKA in these regions, such that the physiological increase in cAMP in the dendrites is insufficient to fully activate the enzyme. This might disrupt LTP induction, particularly late-LTP induction, in the perforant path-DG synapse. For example, four cAMP molecules are required to bind simultaneously to the dimer of R subunits to release active catalytic units, the increase in R2C2 complexes in the somas would have the effect of suppressing activation of PKA because the bound cAMP would be spread over more tetramers, and it would take more total cAMP to activate the same number of catalytic units. However, another reviewer pointed out that since most type II PKA signaling takes place in micro/nanodomains anchored to AKAPs, where cAMP concentrations are locally controlled by AC and PDE activities typically found in these same complexes, as opposed to free in the bulk cytoplasm, the extra Type II PKA in the DG neuron soma/dendrites of AKAP7KO mice is not likely to inhibit normal regulation of AKAP-anchored Type II PKA signaling in those locations. With AKAP anchoring sites likely already being fully occupied, the extra pool of Type II PKA will most likely be cytosolic and primarily found outside these AKAP-organized micro/nanodomains. Signaling by Type I PKA, either unanchored in the cytoplasm (where RI is usually more abundant than RII) or anchored to RI preferring AKAPs, is also not likely to be inhibited by competition for cAMP binding from extra cytosolic type II PKA, both because RI has a higher cAMP affinity than RII and RII is not likely to displace RI from the few known RI preferring AKAPs, which bind RI with greater affinity than RII. This is in contrast to the majority of other AKAPs that are RII preferring, like AKAP7. If the authors agree with this reasoning, the issue should be addressed in the discussion, along with the possibility that a change in plasticity at the DGC dendritic or somal synapses could contribute to the behavioral effect (Although this is less likely than the direct explanation.).*

The issue raised about the possible effect of the increased PKA in the cell body and dendrites when AKAP7 is knocked out is a reasonable one and we have added a section to the Discussion addressing this possibility. We agree with the general discussion of the reviewers but can not rule out a possible effect on behavior caused by this redistribution.

*3) For the two Figure 4, and for the behavioral studies, it would be worthwhile to calculate a measure of effect size, such as Cohen's d, for differences between WT and KO's. The effects are significant, but small. Thus, a statement of Cohen's d, which should be larger than about 0.4, would give a more complete statistical picture of the data. In addition, the Figure 7 comparisons should be against 50% chance to make sure the "successful" group tests are indeed better than 50%. Group comparisons are not enough. Also, the authors should add, in both Figure 5 and Figure 7,a genotype X test ANOVA comparison (e.g., testing% correct if the authors gave more than one test per mouse in Figure 7).*

We have added a measure of effect size (Cohen's d) to the ephys data in Figure 4, Figure 4—figure supplement 1, and Figure 6 (in the figure legends).

In Figure 7 we have added the complete data for the entire 5 day DNMP trial showing that WT and HET mice learn the most difficult 2-arm task and perform statistically better than 50% at days 4-5. The KO and DG-KO differ significantly from their control groups (WT and HET, respectively) by two-way ANOVA over the entire 5 day trial (details in the new Figure 7 legend). We also show the combined data for the test days (4-5) demonstrating a significant difference from respective control groups. We have added Cohen's d values to Figure 7 as well.

*4) Although in Figure 2 it certainly appears that AKAP7 staining in WT mice overlaps more with MF axonal and presynaptic markers (NF-M; ZNT3; SYP) than dendritic and postsynaptic markers (MAP2; PSD-95), adding some quantification of relative AKAP7 co-localization with these markers is needed to support these claims. Likewise for Figure 3, some co-localization quantification should be provided to support the authors' statement that AKAP7 KO results in substantial loss of PKA-Calpha co-localization with NF-M.*

We have added colocalization values to Figure 2 using Image J and Coloc2 as described in the methods and in the figure legend. We note that the markers used to show colocalization were either axonal or terminal specific and AKAP7 is in both compartments giving only partial colocalization shown as intensity correlation quotients (ICQs). In Figure 3 we also analyzed colocalization between NF-M and PKA C subunit using the same method.